# Length-Controlled AlpacaEval:
# A Simple Way to Debias Automatic Evaluators

**Yann Dubois & Percy Liang & Tatsunori B. Hashimoto**
Stanford University
{yanndubs,pliang,thashim}@stanford.edu

## Abstract

LLM-based auto-annotators have become a key component of the LLM development process due to their cost-effectiveness and scalability compared to human-based evaluation. However, these auto-annotators can introduce complex biases that are hard to remove. Even simple, known confounders such as preference for longer outputs remain in existing automated evaluation metrics. We propose a simple regression analysis approach for controlling biases in auto-evaluations. As a real case study, we focus on reducing the length bias of AlpacaEval, a fast and affordable benchmark for chat LLMs that uses LLMs to estimate response quality. Despite being highly correlated with human preferences, AlpacaEval is known to favor models that generate longer outputs. We introduce a length-controlled AlpacaEval that aims to answer the counterfactual question: "What would the preference be if the model's and baseline's output had the same length?" To achieve this, we first fit a generalized linear model to predict the biased output of interest (auto-annotator preferences) based on the mediators we want to control for (length difference) and other relevant features. We then obtain length-controlled preferences by predicting preferences while conditioning the GLM with a zero difference in lengths. Length-controlling not only improves the robustness of the metric to manipulations in model verbosity, we also find that it increases the Spearman correlation with LMSYS' Chatbot Arena from 0.94 to 0.98. We release the code and resulting leaderboard.

## 1 Introduction

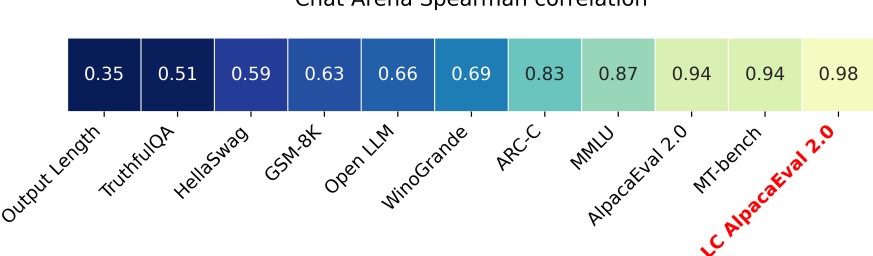

Chat Arena Spearman correlation

Figure 1: Length-controlled AlpacaEval increases correlation with Chatbot Arena from 0.94 to 0.98. It is currently the benchmark with the highest correlation with Chatbot Arena.

Developing and improving NLP systems requires reliable, low-cost evaluations that can quantify progress. In closed-ended tasks, such as multiple-choice QA, such evaluations are straightforward to implement and trust (Novikova et al., 2017; Yeh et al., 2021). However, such evaluations cannot be applied to extremely open-ended settings such as instruction following for language models. Even neural reference-based evaluation metrics such as BERTscore (Zhang* et al., 2020) face challenges in those settings due to the difficulty of collecting a diverse set of references that can cover the space of valid outputs.

Recently, there has been a push toward reference-free evaluation methods that leverage high-performance LLMs, e.g. AlpacaEval (Li et al., 2023), MTBench (Zheng et al., 2023), and WildBench (Lin et al., 2024). While these approaches show a high correlation with human annotators, they often do so by exploiting spurious correlations such as the length of the output, the presence of lists, or various position biases Li et al. (2023); Zheng et al. (2023); Koo et al. (2023); Wang et al. (2023); Wu & Aji (2023).

Creating a way to debias automated evaluation metrics would be highly valuable – it would address the major drawback of LLM-based reference-free evaluations, and enable low-cost, accurate evaluations for developing NLP systems in many open-ended settings. Our work focuses on this challenge of taking an existing automated evaluation metric (e.g. AlpacaEval) and a suspected spurious correlate (e.g. length) and producing a debiased metric.

We propose a simple, interpretable debiasing strategy for automated evaluation metrics based on basic, regression-based adjustments for observational causal inference. We view spurious correlates – such as the length of the response – as undesirable mediators Vander-Weele (2015) in a causal graph and use regression-based causal inference Hernán & Robins (2010) techniques to provide simple adjustments to automated evaluations that control for any suspected spurious correlation.

Applying this approach to the popular AlpacaEval benchmark, we show that controlling for length has significant positive effects on automated evaluation. We find that it is more correlated on average with LMSYS' Chatbot Arena (Zheng et al., 2023) than both (length-uncontrolled) AlpacaEval and MT-bench, and that it is significantly more robust to gaming the evaluation by increasing the verbosity of the output.

Our contributions are the following:

- We propose a simple regression-based debiasing approach for automated evaluation that satisfies several desirable properties for an automatic evaluation metric.
- We apply the approach to AlpacaEval, producing AlpacaEval-LC that is more robust to length-based spurious correlates.
- We show that AlpacaEval-LC correlates better with the human evaluations of model rankings (Chatbot Arena) Fig. 1.

## 2 Background and Problem Setting

Our work relates to both the classic literature on reference-free evaluations, as well as more recent developments in automated and human evaluation of chatbots. We describe some of these relevant works below, with some additional exposition on the details of AlpacaEval, which we study more closely in our debiasing experiments.

**Reference-free evaluation metrics**  Reference-free evaluation metrics have a long history, including classic methods (Louis & Nenkova, 2013) and more recent neural supervised learning methods (Kryscinski et al., 2020; Sinha et al., 2020; Goyal & Durrett, 2020). While this latter class of algorithms has become sufficiently accurate that they match inter-annotator agreement rates, other works have shown that such measurements are heavily confounded by spurious correlations such as perplexity and length (Durmus et al., 2022). Recently, there has been a push to leverage LLMs as a zero-shot, reference-free evaluation measure (Zheng et al., 2023; Dubois et al., 2023; Li et al., 2023; Lin et al., 2024). In the chatbot setting, two well-known such metrics are AlpacaEval and MT-bench, both of which query an LM (based upon GPT4) to attempt to assess the quality of a weaker LM's output.

**AlpacaEval**  AlpacaEval is an LLM-based automated evaluation metric – it operates on a fixed set of 805 instructions chosen to be representative of user interactions on the Alpaca web demo. For each instruction, both a baseline model $b$ (currently GPT-4 turbo) and the evaluated model $m$ produce responses. A GPT-4 turbo-based evaluator then compares the responses head-to-head and outputs the probability of preferring the evaluated model. A win rate is then computed as the expected probability that the auto-evaluator prefers the

evaluated model's output on the 805 instructions. This win rate serves as a performance measure of the evaluated LM chatbot.

Originally, AlpacaEval was designed as a development metric for the Alpaca chatbot (Taori et al., 2023) and AlpacaFarm simulator (Dubois et al., 2023). The metric was designed to control for certain biases, such as the order in which the model and baseline were presented, by randomizing their sequence. However, other factors like length and style effects were not controlled for, as the authors found that humans had similar biases on the analyzed data. Subsequent use of AlpacaEval as a leaderboard revealed that these uncontrolled biases could be significantly gamed by AI systems in ways that human biases couldn't.

Important to our work is that AlpacaEval has several interpretable properties. As a win rate, its values are in $[0\%, 100\%]$, it has a symmetry to baseline swaps, i.e, $\text{AlpacaEval}(b, m) = 100\% - \text{AlpacaEval}(m, b)$, i.e., and comparing a baseline to itself is $\text{AlpacaEval}(b, b) = 50\%$ Any posthoc correction to AlpacaEval should maintain these properties, alongside the usual desiderata of being low-cost, accurate, and robust.

**Chat Arena**   The automated approach to pairwise evaluation in AlpacaEval can be viewed as a low-cost approximation to the Chatbot Arena Zheng et al. (2023), which aims to build real-world human evaluations through live interactions. The approach in Chatbot Arena is that users are presented with a pair of anonymized LMs, and they can send an instruction to both LMs simultaneously. The user receives responses from both LMs and rates the response that is of higher quality. At the end, the head-to-head comparisons are converted to an Elo rating Elo & Sloan (1978) which serves as the model score. As a reminder, the difference in Elo ratings between two players can be converted to a win rate, and vice versa.

This approach has many desirable properties – it is driven by real users and the dynamic nature of the instructions makes it hard to saturate this benchmark. However, this metric cannot be used for model development due to the cost of running many live human evaluations.

In the remainder of this work, we will treat the Chatbot Arena as a silver standard that we wish to approximate. Although Chatbot Arena likely still contains biases (e.g. internet users may focus on surface features rather than "hard to measure" capabilities such as factuality), it represents the largest and most ecologically valid human evaluation process today.

**Problem statement**   We define the problem of pairwise evaluation of a language model in the following way. Given an instruction $x$ sampled from a distribution $p(x)$, a baseline model generates a response $z_b$ and the evaluated model generates a response $z_m$. A human annotator then produces a preference $y \in \{m, b\}$ indicating the model with the better response. An automated surrogate such as AlpacaEval is a (potentially randomized) predictor $f(z_b, z_m, x)$ that aims to approximate the corresponding human label $p(y = m | z_b, z_m, x)$. AlpacaEval's current win rate is $\text{winrate}(m, b) = 100 \cdot \mathbb{E}_x[f(z_b, z_m, x)]$, i.e., the expected predicted preference of human annotators for the model response over the baseline's response.

## 3   Length-Controlled AlpacaEval

A major challenge in building automated evaluations $f(z_b, z_m, x)$ is the *spurious correlations* problem. Specifically, consider a simple example in which there is a spurious correlate $c$ (e.g. length) such that heavily relying upon $c$ can be predictive of the human label $y$. The confounder $c$ is initially predictive of $y$ but becomes less predictive as model builders explicitly begin to optimize against the metric. Adopting this causal view, we ask our motivating question

> **What would the AlpacaEval metric be, if the outputs of all models had the same length as those of the baseline?**

Our goal in this section will be to operationalize this into a simple regression-based estimator. To be precise, we hypothesize that automated evaluation measures $f$ such as AlpacaEval return their quality estimates through a combination of *direct* effects that measure the quality of the model response and *indirect* effects that are mediated by spurious variables such as

the length of outputs. The goal of controlling for the spurious correlates is thus equivalent to controlling these indirect effects. See Fig. 2 for a visual representation.

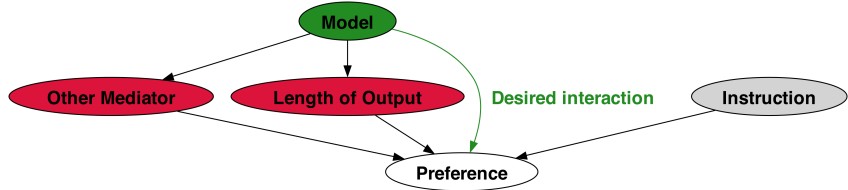

Figure 2: Length-Controlled AlpacaEval predicts the direct effect of the model (green) on the auto-annotators' preference (white) when controlling for undesirable mediators (red) and other useful features (gray).

This abstraction leads us to a simple approach for bias correction, inspired by methods for estimating Controlled Direct Effect (VanderWeele, 2010), for which simple Generalized Linear Models (GLMs) can provide reasonable estimates.

**Length control via regression** Our approach will be to estimate the contribution of 3 different components to the AlpacaEval quality judgment:

- **Model identity** Whether an output comes from the baseline model $b$ or the evaluated model $m$ should impact the probability that an output wins the pairwise comparison.
- **Length of output** The length of output is known to affect both human and model judgments of output quality (Dubois et al., 2023; Singhal et al., 2023), and so we expect this to also affect the win probability.
- **Instruction difficulty** Models do not perform uniformly over instructions: the preference of humans will generally depend on the instruction. For example, the baseline might be much better for coding tasks than any other tasks. For every instruction we thus want to model the difficulty of that task for the baseline. Note that the (baseline) "instruction difficulty" is not caused by "model" but conditioning on it can enhance the precision of estimates in regression analysis by reducing unexplained variability Pearl (2009).

We can obtain length corrected AlpacaEval score in two steps: (i) first, we can fit a model to these three attributes, and (ii) then we zero out the "length of output" term to obtain counterfactual estimates of AlpacaEval win rate.

**The regression model** Motivated by the previous discussion, we will model the AlpacaEval predictions $f(z_m, z_b, x)$ with a logistic regression that has 3 terms: model, length, and instruction. We will first present the overall regression formula, explain the details of the featurization, and then describe some naturally appealing properties of our featurization.

$$q_{\theta,\phi,\psi}(y = m | z_m, z_b, x) :=$$

$$\text{logistic}\left( \underbrace{\theta_m - \theta_b}_{\text{Model}} + \underbrace{\phi_{m,b} \cdot \tanh\left( \frac{\text{len}(z_m) - \text{len}(z_b)}{\text{std}(\text{len}(z_m) - \text{len}(z_b))} \right)}_{\text{Length}} + \underbrace{(\psi_m - \psi_b)\gamma_x}_{\text{Instruction}} \right)$$

(1)

The model and instruction terms are straightforward – they can be viewed as the log-linear contribution of the model $(m, b)$ and each instruction's difficulty $(\gamma)$ on the baseline win rate. The length term is linear in a *normalized* length feature, where the normalizer standardizes the length to have unit variance and transforms this via a tanh, as differences in lengths should have strong diminishing returns on the log odds.

Importantly, this formula fulfills the identity property, i.e., $q(y = m \mid z_b, z_b, x) = 0.5$, and symmetry property, i.e, $q(y = m \mid z_m, z_b, x) = 1.0 - q(y = b \mid z_b, z_m, x)$ of the original

win rate. Identity holds as the length term is zero due to having no difference in length, while the other two terms are zero as the coefficients are identical. For symmetry note that $\text{logistic}(x) = 1 - \text{logistic}(-x)$, and it is clear that swapping $m$ and $b$ flips the sign of the model and instruction terms. For the length term, the same is true as flipping $m$ and $b$ negates the length difference, and tanh is an odd function. More generally any additive term that is antisymmetric and centered around 0 would satisfy the desired properties.

**Obtaining length corrected (LC) win rate**    Using the model from Eq. (1) we can answer the counterfactual question of what the automatic evaluation $f$ might be if the length of the evaluated model matched that of the base model, i.e., $\text{len}(z_m) = \text{len}(z_b)$. In this case, the second, length term becomes zero and we obtain the length corrected win rate estimate as

$$\text{winrate}^{LC}(m, b) = 100 \cdot \mathbb{E}_x\left[\text{logistic}\left(\theta_m - \theta_b + (\psi_m - \psi_b)\gamma_x\right)\right], \qquad (2)$$

i.e., we remove the length term from the regression and compute the implied win rate.

**Training**    Training of the regression is simple and uses off-the-shelf libraries for fitting generalized linear models. Since our GLM uses a logit link function, we fit the model in Eq 1 using the cross-entropy loss $\mathcal{L}(\theta, \phi, \psi) = \mathbb{E}_{p(y|z_m, z_b, x)p(z_m, z_b, x)}\left[q_{\theta, \phi, \psi}(y|z_m, z_b, x)\right]$.

AlpacaEval's leaderboard uses a constant baseline $b$, so without loss of generality we can drop $\theta_b, \psi_b$, which can be absorbed into the corresponding parameters for $m$. In total, for a leaderboard with $M$ models and $N$ instructions, our GLM contains $3M + N$ parameters to be estimated from $MN$ examples ($\theta_m, \phi_{m,b}, \psi_m$ for each model, $\gamma_x$ for each instruction). This will be overdetermined when $M$ and $N$ are both large, as in the case of AlpacaEval ($M > 128$, $N = 805$). However, to ensure our procedure is robust even for small $N$ and $M$, we use 5-fold cross-validation with $L_2$ regularization on weights to avoid potential overfitting.

The one complexity of our regression is that the instruction difficulty term $\gamma_x$ is shared across models, and so we estimate this separately by first fitting a joint regression across all models with the $\psi_m - \psi_b$ term fixed to one and using the estimated $\gamma_x$ from this regression.

For the remaining regression coefficients, we simply fit $\theta$, $\phi$, and $\psi$ on the AlpacaEval predictions for each model separately, re-using the already estimated $\gamma_x$ as these do not depend on the model being evaluated. Fitting models separately is important as it implies that previously computed metrics won't change when adding a new model to the leaderboard.

Finally, we have added an additional weak regularization on $\phi_{m,b}$ to prevent an adversary from performing attacks that intentionally truncate sequences that a model performs poorly on. In this case, the poor performance of the model would be perfectly correlated with the short length, and the model builder would be able to exploit the length corrections to boost the performance of the model. Adding a regularization term makes it so that any model performance issues would be explained by the model terms first, and then any residual effects would be captured by the length effects, as intended. The regularization is weak enough that we empirically found it to not affect non-adversarial models.

# 4   Results

We apply our approach to all of AlpacaEval, as this benchmark has known length confounders, contains a large set of pre-computed LLM-based pairwise comparisons ($> 120$ models, 805 instructions), and is widely used by the research community. We evaluate our approach on several measures of interest:

- **Decreasing length gameability**: We call a metric length gameable if simply prompting the model to be more or less verbose significantly affects the metric outcome. Ideally, length gameability should be low for two reasons. First, we would like evaluations that prioritize the content rather than the style of the answer. Second, the benchmark should not be too dependent on the prompting strategy as users usually think of evaluations of models rather than the entire system, which includes the prompt.

- **Correlation with chatbot arena**: If our gameability and robustness metrics represent better capturing human preference, we should improve our correlation with chatbot arena. We measure Spearman rather than Pearson correlation as probabilities are log-linearly correlated with ELO ratings, rather than linearly.
- **Robustness and interpretability**: Our corrected metric should be robust to simple adversarial attacks such as truncation, and be interpretable to users as a win rate.

We show that AlpacaEval-LC fulfills all these goals. We release the code for all experiments.

| | AlpacaEval | | | Length-controlled AlpacaEval | | |
|---|---|---|---|---|---|---|
| | concise | standard | verbose | concise | standard | verbose |
| gpt4_1106_preview | 22.9 | 50.0 | 64.3 | 41.9 | 50.0 | 51.6 |
| Mixtral-8x7B-Instruct-v0.1 | 13.7 | 18.3 | 24.6 | 23.0 | 23.7 | 23.2 |
| gpt4_0613 | 9.4 | 15.8 | 23.2 | 21.6 | 30.2 | 33.8 |
| claude-2.1 | 9.2 | 15.7 | 24.4 | 18.2 | 25.3 | 30.3 |
| gpt-3.5-turbo-1106 | 7.4 | 9.2 | 12.8 | 15.8 | 19.3 | 22.0 |
| alpaca-7b | 2.0 | 2.6 | 2.9 | 4.5 | 5.9 | 6.8 |

Figure 3: Length-controlled AlpacaEval decreases the sensitivity to prompting the evaluated model for more concise or verbose outputs.

## 4.1 AlpacaEval-LC decreases length gameability

A good evaluation metric should not be so sensitive to length that prompting for longer or shorter responses completely changes the metric. To measure gameability we prompted different models to "Answer with as much detail as possible." (verbose) or "Be as concise as possible while still providing all the necessary information to answer the question." (concise).

Figure 3 shows that AlpacaEval is highly length gameable. The baseline model (gpt4_1106_preview) fluctuates from 22.9% to 64.3% by varying the verbosity instruction in the prompt. Even worse, significant gains are possible by asking weaker models to be verbose, as seen with Claude-2.1.

In contrast, the length-controlled AlpacaEval has significantly lower gameability (gpt4_1106_preview's win rates now only fluctuate from 41.9% to 51.6%), and rankings are generally stable to verbosity prompts. Quantitatively, the normalized standard deviation across the three verbosity prompts decreases from 25% to 10% from the length control.

## 4.2 AlpacaEval-LC increases correlation with Chatbot Arena to 0.98

Our prior experiments demonstrate that length control reduces the high sensitivity to length in AlpacaEval. However, our goal is not simply to make metrics that are less sensitive to length, but to produce metrics that are overall more representative of human judgments.

Figure 1 shows that controlling for length increased the Spearman correlation with Chat Arena from 0.94 to 0.98. Of existing benchmarks, this difference is significant enough to make the length-corrected version of AlpacaEval the metric with the highest correlation with Chat Arena which we are aware of. Correlations are computed on every benchmark that evaluates at least 25 models from the Chatbot Arena. AlpacaEval and AlpacaEval-LC have 38 such models, MT bench has 34. The bootstrap p-value comparing the correlation with AlapcaEval's correlation is 0.07 and 0.06 compared to MT-bench.

**Length control generally improves the rankings of proprietary models** Figure 4 shows leaderboard changes due to our length control approach. We see that proprietary models,

which often generate shorter responses, perform much better on AlpacaEval-LC, and the biggest rank losses are in open-source models that have gone through the RLHF process Ouyang et al. (2022). Given that AlpacaEval is a potential optimization target for open-source language models, these results are consistent with the hypothesis that existing open models had exploited the length bias of AlpacaEval.

| | Length | Win Rate | New Win Rate | Win Rate Gain | Rank Gain |
|---|---|---|---|---|---|
| gpt4_1106_preview | 2049 | 50.0 | 50.0 | 0.0 | 0 |
| claude-3-opus-20240229 | 1388 | 29.0 | 40.4 | 11.4 | 5 |
| gpt4 | 1365 | 23.6 | 38.1 | 14.6 | 8 |
| Qwen1.5-72B-Chat | 1549 | 26.5 | 36.6 | 10.1 | 5 |
| gpt4_0314 | 1371 | 22.1 | 35.3 | 13.2 | 7 |
| claude-3-sonnet-20240229 | 1420 | 25.6 | 34.9 | 9.3 | 4 |
| mistral-large-2402 | 1362 | 21.4 | 32.7 | 11.2 | 10 |
| Samba-CoE-v0.2-best-of-16 | 1578 | 27.0 | 31.5 | 4.5 | 0 |
| gpt4_0613 | 1140 | 15.8 | 30.2 | 14.4 | 20 |
| Snorkel-Mistral-PairRM-DPO-best-of-16 | 2616 | 34.9 | 30.0 | -4.9 | -8 |
| Contextual-KTO-Mistral-PairRM | 2521 | 33.2 | 29.7 | -3.5 | -8 |
| pairrm-Yi-34B-Chat | 2195 | 31.2 | 28.8 | -2.4 | -8 |
| mistral-medium | 1500 | 21.9 | 28.6 | 6.8 | 0 |
| claude-2 | 1069 | 17.2 | 28.2 | 11.0 | 9 |
| Samba-CoE-v0.2 | 1469 | 21.8 | 27.6 | 5.8 | -1 |
| falcon-7b-instruct | 478 | 2.1 | 4.0 | 1.9 | 2 |
| oasst-sft-pythia-12b | 726 | 1.8 | 3.3 | 1.5 | 3 |
| guanaco-13b | 1774 | 3.5 | 3.0 | -0.5 | -12 |
| guanaco-7b | 1364 | 2.9 | 2.9 | -0.0 | -7 |
| baichuan-13b-chat | 1727 | 2.0 | 2.1 | 0.1 | -1 |

Figure 4: Closed source models often perform better (red) on length-controlled AlpacaEval as they are often shorter. The first column shows the length of outputs. The 4th and 5th columns, respectively, show the win rates and rank gains due to LC. The table shows the top 15 and bottom 5 systems in the leaderboard.

## 4.3 AlpacaEval-LC is interpretable and robust

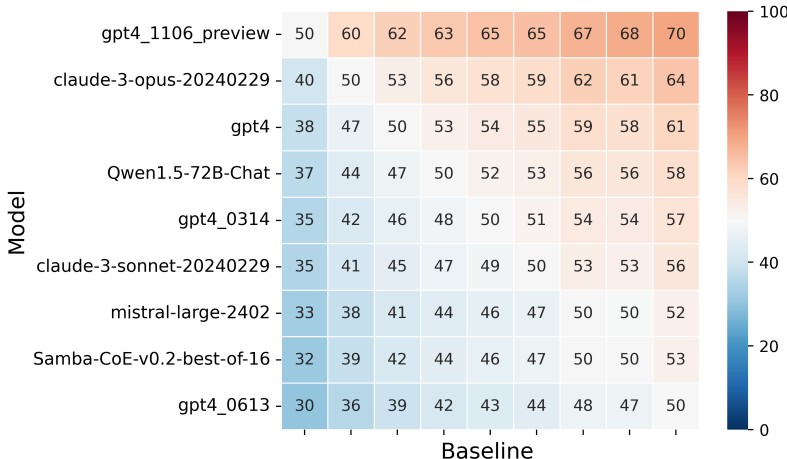

Figure 5: With our GLM we can predict what the win rate would be if the baseline was any other model from the leaderboard. The $x$-axis shows the baseline, and the $y$-axis the model. Both are in the same order.

**Regularization makes AlpacaEval-LC robust to truncation**  One potential issue with simple bias corrections is that they may be gamed through white-box adversarial attacks, e.g., postprocessing the outputs of models to make them look better on AlpacaEval-LC. One example of such an attack is to truncate all outputs to a few characters, besides those that are much better and around the same length as the baseline. A naive GLM fitted on such outputs should naturally predict very high win rates in the counterfactual world where outputs have the same length as the baseline. Indeed, when doing such post-processing to GPT-4 outputs, win rates increase from 3.7 (AlpacaEval 2.0) to 25.9 (AlpacaEval-LC, no regularization). To mitigate such adversarial attacks, our approach includes a regularization term on $\phi_{m,b}$. This decreases the gamed win rate to 12.2 (AlpacaEval-LC with regularization) while having an imperceptible impact on non-adversarial models.

**Interpretability as a win rate**  Figure 5 shows that LC win rates can be interpreted similarly to raw win rates. In particular, the baseline always has a win rate of 50% and winrate$(m, b) = 100\% -$ winrate$(b, m) \in [0\%, 100\%]$. This seems very natural but wouldn't hold for most length-correction methods, such as normalizing by length.

More interestingly, a nice property of our GLM is that once we fit the weights for one baseline, we can predict the win rate between any pair of models on the leaderboard. As a result, we can predict the leaderboard for any other baseline as seen in Fig. 5.

## 4.4  Comparisons to baselines for length control

|  | Arena Correlation (↑) | Gameability (↓) | Adversarial Win Rate Gain (↓) |
|---|---|---|---|
| **Win Rate** | 0.94 | 26% | 0.0 |
| **Length-Controlled Win Rate** | 0.98 | 10% | 8.5 |
| **Length-Normalized Win Rate** | 0.96 | 15% | 3.6 |
| **Length-Balanced Win Rate** | 0.95 | 15% | 40.8 |

Figure 6: Length-controlled win rate has the best Arena Correlation and gameability from considered methods, while still being relatively robust to adversarial attacks.

Let's now briefly discuss two other potential family length-correction methods that have been proposed in the community Duong (2024); Galambosi (2024); Teortaxes (2024).

**Length-balanced win rate**  Another common way to control some covariates is through stratification. One potential metric, dubbed length-balanced (LB) win rate, would thus be to compute the average win rate stratified on examples where the model outputs are (1) longer and (2) shorter than the baseline Duong (2024). LB satisfies many of the desiderata of length control but has one main downside: robustness.

In particular, stratification relies upon having enough samples within each stratum, otherwise the estimates may rapidly become unstable. This can increase variance, e.g., if one model is naturally longer than another, but can also introduce adversarial vulnerabilities.

The first and last rows in Fig. 6 show that length-balanced win rates improve both the length gameability (measured by the normalized standard deviation of win rate across concise/standard/verbose prompts) and the Chatbot arena correlation. However, this approach is strictly dominated by our length-controlled method – in arena correlation, gameability, and adversarial win rate gains from truncating bad GPT-4 outputs as discussed in Section 4.3.

**Length-normalized win rate**  Another option Galambosi (2024); Teortaxes (2024) is to directly normalize the win rate by a function of the length of the model's and baseline's output. We have tried several variations on normalization (e.g. directly dividing by lengths, logistic function of lengths, etc). In our experiments, the function that performed best was dividing the raw win rate by a temperature-scaled logistic function of the average difference of lengths. We call this metric length-normalized (LN) win rate.

Figure 6 shows that this simple LN win rate performs surprisingly well on many of the metrics. We chose to present and implement the length-controlled (LC) win rate, as it is more principled (as an estimate of the direct effect), interpretable (as a win rate), and performs slightly better on all quantitative metrics except adversarial gameability.

## 5 Discussion

| | Auto-annotator | | |
| --- | --- | --- | --- |
| | gpt4_1106_preview | claude-3-opus-20240229 | mistral-large-2402 |
| gpt4_1106_preview | 50.0 | 50.0 | 50.0 |
| claude-3-opus-20240229 | 40.4 | 43.3 | 47.5 |
| mistral-large-2402 | 32.7 | 28.2 | 45.5 |
| gpt4_0613 | 30.2 | 20.5 | 34.3 |
| gpt-3.5-turbo-1106 | 19.3 | 16.7 | 28.9 |

Figure 7: Length-controlled win rate has the best Arena Correlation and gameability from considered methods, while still being relatively robust to adversarial attacks.

**Other biases**   Length is a well-known bias of automated evaluators of chatbot LLMs but several others have been noted, including a bias of models towards their own outputs Zheng et al. (2023), or presence of lists Dubois et al. (2023). While we focus on a more detailed study of length biases here, we note that the same approaches can be applied to other biases by representing them as additional features in the logistic regression.

Additionally, our preliminary explorations of self-annotator biases shows that the effect exists but is often smaller than general model differences. Fig. 7 shows that the ranking of considered models does not change when using different annotators. In particular, Claude 3 Opus prefers GPT4 Preview, and Mistral Large prefers the former two than itself.

**Length-controlling in RLHF**   Our work is closely related to the recent work that aims to debias (implicit or explicit) reward models used to finetune LLMs with RLHF (Singhal et al. (2023)). For example Shen et al. (2023); Chen et al. (2024) try to train a reward model that is uncorrelated to length by making it predict the length at the same time as the reward and disentangle the two. Park et al. (2024) extends this intuition to the case of implicit reward models. This type of debiasing would not work out-of-the-box in typical auto-evaluation settings, e.g. AlpacaEval, which uses closed source LLM as judges rather than training a reward model. Our post-hoc debiasing could however be used in the RLHF setting, and we encourage future work to look into that.

**Limitations**   Firstly, we only tested our proposed debiasing mechanism on the AlapcaEval benchmark, which uses a set of relatively simple English instructions and a particular prompt for the LLM judge. Secondly, AlpacaEval-LC is based on the simplifying assumption that you would like to compare the model and the baseline as if they had the same length. Finally, we do not (aim to) solve any other issues associated with the use of an LLM-judge Zheng et al. (2023). Despite these limitations, we show that the correlation with chat arena increases significantly, which suggests that AlpacaEval-LC takes a step in the right direction.

**Conclusion**   We propose a simple method for mitigating the length bias of LLM-based automatic evaluations, specifically, AlpacaEval. The procedure consists of fitting a generalized linear model to predict the auto-evaluators preferences, conditioned on the length of the models' output. We then get the length-controlled preference by predicting what the auto-evaluator would have preferred if the model's output and the baseline's output had the same length. We show that the resulting length-controlled AlpacaEval, has higher correlations with humans, has much less length bias, and is robust (hard to game).

## Acknowledgments

First, we note that Balázs Galambosi is an author on this paper as shown on the Arxiv version of the paper, but we can't update the author list here. We thank Xuechen Li, Rohan Taori, and Tianyi Zhang for help maintaining AlpacaEval. We thank Viet Hoang Tran Duong for suggesting to consider length-balanced win rates. We thank the Twitter ML community for emphasizing the need of length-controlled autoevaluations. We thank the community for all the 100+ models they added to AlpacaEval. We thank OpenAI and Together AI for API credits to generate outputs and evaluate models. We gratefully acknowledge the support of an Open Philanthropy Project Award. Tatsunori Hashimoto is supported by the Tianqiao and Chrissy Chen Institute. Yann Dubois is supported by a Knights-Hennessy Scholarship.

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
