# OpenReview forum: "Length-Controlled AlpacaEval: A Simple Debiasing of Automatic Evaluators"
_colmweb.org/COLM/2024/Conference — COLM_

### Official Review · Reviewer_suS5 · 2024-05-10

**Rating:** 6
**Confidence:** 3
**Ethics Flag:** 1

**Summary:**

This paper describes a methodology for controlling for length bias in the automatic LLM evaluation methodology AlpacaEval. The method involves decomposing AlpacaEval winrates into a logistic regression problem with an explicit output length parameter. The resulting corrected metric has good Chatbot Arena correlation and shows resilience to gameability. It's a simple method that controls for a simple confounder.

**Questions To Authors:**

Length bias is a highly interesting phenomenon, likely rooted to a strong correlation between length and quality. I wonder if zero-ing this out is actually the "correct" way to go about this?


Writing suggestions:

- Section 4: "Ideally, length gameability is low" -> "Ideally, length gameability should be low"

- Section 4.1: "changes the me" -> "changes the metric?"

- Section 4.2: "We see that proprietary models..." this point is not immediately clear to me. Proprietary models would also have undergone RLHF, and also may have been optimized for widely used evals. It could be noted which models here are considered "proprietary" to make a distinction. I think I understand the general sentiment, but think it would be more appropriate to talk about it as a matter of training data rather than proprietary vs open-source.

- Section 4.3: "Regulatization makes LCAE" LCAE -> AE-LC?

- Section 4.3: "imperceptible impact" could this be elaborated on?

- minor point, but "winrate" "win rate" and "win-rate" are used throughout the paper.


=== comments after response ===

Thank you for the response. I'm glad to hear about the updates.
I believe the paper has a narrow, but solid contribution. I will keep my scores.

**Reasons To Accept:**

Good utility:

Automatic evaluation is a valuable topic. Assuming Chatbot Arena as a gold standard, creating automatic evaluation metrics that best mimics it without human crowdsourcing is useful. The proposed approach in the paper results in a very high correlation. The paper does a good job of convincing the necessity of length correction.

Comparisons with alternative methods:

The methodology is simple, but comparisons are made with other methods of length correction methods. Analysis shows that the proposed method has strong gameability resistance and best Arena correlation.

**Reasons To Reject:**

Limited scope:

The scope of the paper is limited to improving AlpacaEval's winrate-based methodology. There are no experiments to assess the length-controlling method for other automatic evaluators. It would have been interesting to adopt the method to investigate length bias of other metrics. Given that the paper solely focuses on AlpacaEval, the utility of this method inherits much of the limitations of AlpacaEval itself (LLM as judge may be inaccurate to evaluate stronger models or harder queries).

---

> ### Author Rebuttal · Authors · 2024-05-26
>
> Thank you for your constructive review and detailed suggestions. We've updated the manuscript to incorporate your suggestions and talk about the limitations that you've highlighted.
>
> > Length bias is a highly interesting phenomenon, likely rooted in a strong correlation between length and quality. I wonder if zeroing this out is actually the "correct" way to go about this?
>
> This is a great question and we hope our work spawns more work in that domain to consider what should be done concerning length bias. We think that this is the right way of answering our motivating question (section 3):
>
> "What would the AlpacaEval metric be, if the outputs of all models had the same length as those of the baseline?".
>
> But we agree that the question itself may not be what you want to answer. We believe that the increase in correlation with ChatBot Arena compared to no length correction, and to other methods (see Figure 6) shows that this is a sensible step in the direction of controlling for length bias. Still, we believe that much more research has to be done to show what is the ideal way of considering length in our evaluation benchmarks.
>
> > Given that the paper solely focuses on AlpacaEval, the utility of this method inherits much of the limitations of AlpacaEval itself [...]
>
> Yes, we completely agree and added a limitation section to the updated manuscript that highlights the main limitations of the method including the ones you mentioned.

---

> > ### Comment · Reviewer_suS5 · 2024-06-05
> >
> > Thank you for the response. I'm glad to hear about the updates.

---

### Official Review · Reviewer_39UA · 2024-05-10

**Rating:** 8
**Confidence:** 4
**Ethics Flag:** 1

**Summary:**

This paper aims at improving AlpacaEval (a framework that uses LLMs to judge instruction-following models' quality, using the metric of "win rate" compared to a baseline model) by mitigating the length-bias issue: it is well known that LLM judges have verbose biases (prefer longer outputs).

The authors propose to use _all the predicted results of LLM judges and several tested models_ to fit a logistic regression model. The features include (1) the models tested (how strong the model is), (2) the instruction used (how difficult the instruction is), and (3) the length of the response. Then the length-controlled win rate can be calculated by the regression model without using the length feature. Specifically, the authors added a weak regularization term on the weight of the length feature to avoid attacks (by deliberately truncating the response).

The paper has done a thorough analysis on the proposed length-control AlpacaEval. They showed that (1) the new evaluation reduced the chances the model can game the benchmark by being more verbose (through prompting); (2) the correlation with chatbot arena (human evaluation) is the highest among all model-based evaluation; and (3) it is robust to attacks like truncating the model responses. The authors also compared this method to some naive length-controlled evaluation and showed the advantage of the proposed method.

**Questions To Authors:**

How does the number of models affect the performance of the regressor? I did not find a lot of details on this, but I imagined all the experiments are done on dozens of models. What happens if you only have a few models evaluated? (could be a realistic scenario when you have a new benchmark that only a few models are available, or even just two!)

**Reasons To Accept:**

(1) The discussed topic is of high impact and of high value in the language model community (model-based evaluation for instruction-following models). The proposed new evaluation mitigated a significant bias in those evaluations -- length biases.

(2) The evaluation of the method is thorough -- the authors demonstrated, from different perspectives, that length-controlled AlpacaEval is better.

(3) The proposed method is principle and novel compared to some of the existing solutions to the length bias problem.

**Reasons To Reject:**

(1) The paper's writing could be improved. The description of the regression part (the main method) is especially unclear. For example, the authors did not explain $\theta_m$, $\theta_b$, (very easy to be confused with LM parameters), $\psi_m$, and $\gamma_x$ in Equation (1). It is also unclear what data (how many models, how many instructions) the authors used to train the regressor, which could be critical details.

(2) From my understanding, every time there is a new model, the evaluation needs to retrain the regressor, which leads to additional overhead and constantly changing numbers. This could be a limitation of the method.

---

> ### Author Rebuttal · Authors · 2024-05-26
>
> Thank you for your constructive review and great questions. We clarified and highlighted the following answers to your questions in the updated manuscript.
>
> > It is also unclear what data (how many models, how many instructions) [...]
>
> The regression model was trained on the entire open AlpacaEval benchmark at the time of submission. This means 805 instructions and >120 models. We clarified that in the updated manuscript.
> Note that only the (805) instruction parameters are shared across different GLMs. All the other parameters (3 per model) are independent between models. Once the instruction parameters are fitted, our pipeline can be thus seen as training 120 separate GLMs (one per model). Each GLM contains 3 parameters and is trained on 805 examples in less than a second. The entire pipeline thus should work well at different scales of models and instructions.
>
> > From my understanding, every time there is a new model, the evaluation needs to retrain the regressor, which leads to additional overhead and constantly changing numbers.
>
> This is an important concern that we wanted to avoid. As explained in the previous answer and at the end of section 3: once the instruction parameters are fitted we can fit models separately and "Fitting models separately is important as it implies that previously computed metrics won't change when adding a new model to the leaderboard'. In summary, we fit 805 parameters on 805*120 data points once, and for each model, we can then fit each of the 3 parameters of the GLM separately.
>
> > How does the number of models affect the performance of the regressor?
> > What happens if you only have a few models evaluated?
>
> As mentioned above, only the parameters that quantify instruction difficulty are shared, so the question is whether you can evaluate those shared parameters with a few models. First, to put things in perspective, we found that even if you drop the instruction difficulty terms we get 0.96 correlation with ChatBot Arena, so if you only had a few models you would be fine just dropping that term. Second, for every instruction the instruction difficulty is a scalar trained with regularization and cross-validation, we would thus expect to be able to get a useful estimate of it even with a few models being evaluated.

---

> > ### Comment · Reviewer_39UA · 2024-06-05
> > **Ack**
> >
> > Thanks for answering my questions!

---

### Official Review · Reviewer_P3wF · 2024-05-10

**Rating:** 7
**Confidence:** 4
**Ethics Flag:** 1

**Summary:**

This work proposes a method for mitigating the length bias of LLM-based evaluations -- AlpacaEval in this case. They fit a generalized linear model to predict the LLM evaluator's preferences, conditioned on the length of the models’ output. The length-controlled preference is then obtained by predicting what the LLM evaluator would have preferred if the model’s output and the baseline’s output had the same length. LC-AlpacaEval is shown to have a higher correlation with humans, reduced length bias, and is less sensitive to length changes.

**Questions To Authors:**

* Have you considered other confounders covered in related work, such as the length of instructions, specific word choices,  and so on ? Specifically, using Koo et al. (2023)'s taxonomy, the paper addresses Salience bias (length), Egocentric bias (model identity), and an additional challenge of difficulty of instructions. Do you see any other biases that could exist in the framework (e.g. order bias or compassion fade). How could those be addressed, especially when they co-occur with length?
* It would be beneficial to add recommendations for what improvements LLMs can make (especially while collecting human feedback for alignment) for controlling biases such as this one.

**Reasons To Accept:**

* The work proposes a simple and effective approach for controlling biases in LLM-as-judge evaluations.
* Shows strong correlations with Chatbot Arena.
* The paper is well-written, explains the nuances well, and is easy to understand.
* Flexibility  -- once the weights are fitted for one baseline, win rate between any pair of models on the leaderboard can be predicted.
* Offers some interesting insights into open-source LLMs:
> these results are consistent with the hypothesis that existing open models had exploited the length bias of AlpacaEval.

**Reasons To Reject:**

Some crucial details are missing:
* Which data exactly is the regression model trained on?
* Details of models and instructions is missing.
* What were the ranges of M (# of models) and N (# of instructions) used for experimentation?

---

> ### Author Rebuttal · Authors · 2024-05-26
>
> ​​Thank you for your constructive review and great questions. We clarified and highlighted the following answers to your questions in the updated manuscript.
>
> > Which data exactly is the regression model trained on?
>
> > What were the ranges of M (# of models) and N (# of instructions) used for experimentation?
>
> The regression model was trained on the entire open AlpacaEval benchmark at the time of submission (805 instructions and >120 models). We clarified that in the updated manuscript. Note that only the (805) instruction parameters are shared across different GLMs. Once the instruction parameters are fitted, our pipeline can be thus seen as training 120 GLMs (one per model), each on 805 examples and each containing 3 parameters. Fitting each GLM takes less than a second. The entire pipeline thus should work well at different scales of models and instructions.
>
> > Have you considered other confounders covered in related work [...]
>
> Great question! We briefly discussed other confounders in section 5 under ``Other biases'' but did not have the space to properly answer this question. One of the reasons we like the regression analysis approach is that it should be easy to extend it to new confounders by controlling for them (see figure 2). In practice, this would translate to adding a new term/feature in the GLM. To maintain all the properties of win rates discussed in section 4 you would just need to ensure that the added term is antisymmetric and centered around 0. We are thus relatively optimistic about extending our method to new biases.
>
> In terms of which biases to control for, here are three ones we thought about and why we decided not pursue them in this paper:
> - Order bias: LLM preference depends on the order in which the examples were shown. AlpacaEval avoids this issue by randomizing the order across prompts.
> - Egocentric bias: Figure 7 shows that this bias exists but does not impact the ranking between models (for the considered models). Furthermore, egocentric bias mostly impacts one model out of all the models on a benchmark. We thus decided not to control for it.
> - List bias: Figure 9 in the [AlpacaFarm](https://arxiv.org/abs/2305.14387) paper shows that LLMs prefer outputs that contain lists. We think that it would be an interesting bias to consider but we decided to prioritize length since AlpacaFarm showed that humans prefer lists even more than LLMs, and since it’s harder to quantify the amount of lists in an answer.

---

### Official Review · Reviewer_3Bhq · 2024-05-11

**Rating:** 7
**Confidence:** 4
**Ethics Flag:** 1

**Summary:**

This paper proposes a simple regression model to adapt existing LLM-based LLM-chat evaluation metrics to control for different sources of noise in automated evaluations (demonstrating the method by length-adapting the Alpaca metric). They fit a generalised linear model to predict which of two models is better for a given instruction based on the predictions of the existing evaluator (Alpaca), the relative length of the outputs of the two models, and the instruction/prompt given to elicit the output. After fitting this model, they use it for prediction by ignoring the length term. This results in a model which has better correlations with human performance (0.98 vs the previous best of 0.94) and the authors also consider factors like 'gameability'--how easy is it to tweak your outputs (e.g. through prompt manipulation) to get the scores or 'win-rate' you want and demonstrate that their method performs well for at least one measure of gameability.

Overall, this paper seems like a worthwhile contribution for researchers interested in approximating human preferences for English-language chatbot LMs without a desire for finer grained assessements (e.g. of accuracy, factuality, naturalness, style, grammaticality, etc).

**Questions To Authors:**

Requests and observations:
- I don't think the figure on your first page helps with understanding the problem or your solution, and more properly belongs later in the article.
- please update your figures so that the colors are distinguishable in grayscale (e.g. when printed black and white or viewed on an e-ink display) and ensure that all text embedded in the figures is legible (e.g. sometimes there is black text on a dark background that is very difficult to read)
- please convert your tables to native LaTeX so that they can be highlighted while reading.

**Reasons To Accept:**

Clearly articulated model and benefits for metrics used on a popular leaderboard. Reasonable evaluation to support the claims.

**Reasons To Reject:**

The focus on leaderboards and 'overall quality' metrics oversimplifies the complexity of evaluation. The paper is implicitly about evaluating English-language LLMs but never states this explicitly.

---

> ### Author Rebuttal · Authors · 2024-05-26
>
> Thank you for your constructive review. We incorporated your feedback, which we believe improved our manuscript!
>
> In particular, we added a limitation section, which now explicitly states that the paper only considers prompts written in English and that we inherit all the limitations from focusing on a single-metric leaderboard for evaluation.

---

> > ### Comment · Reviewer_3Bhq · 2024-06-03
> >
> > Glad it helped! It would be good to mention the target language in the title or abstract in addition to the discussion in the limitations section, just to help people searching for relevant papers :)

---

### Comment · Area_Chair_BivJ · 2024-06-03
**[Area Chair Comment]: Author Response**

Dear reviewers,

Please take a look at the author's rebuttals and the other reviews for this paper! If the rebuttals addressed your concerns, please let the authors know about this and update your review. If not, please continue to engage with the authors and the other reviewers in the discussion forum.

Thanks!

---

### Decision · Program_Chairs · 2024-07-10

**Decision:**

Accept

**Comment:**

The paper aims to address the length bias issue in LLM-as-a-judge framework, focusing specifically on AlpacaEval. The core idea of the paper is to train a logistic regression model to fit the LLM judge predictions for all tested models using the tested models, instruction difficulty, and length difference with a baseline response as the features. The original alpacaEval winrates are converted into length controlled win-rates by zero-ing out the length difference in the above LR model.

Pros:
Overall, the reviewers agree that the idea is pretty neat, and the experiments clearly show that this reduces the game-ability of the metric by manipulating the generation's lengths. They also report higher correlation with the ChatBot Area, i.e. human evaluation, benchmark.

Minor Cons:
The paper only validates their approach on one LLM-as-as-judge framework. But, given the popularity of AlpacaEval, this is a minor criticism.
The authors should also incorporate the reviewer comments about improving the presentation of the paper.